# BRAIN-LIKE FUNCTIONAL ORGANIZATION WITHIN LARGE LANGUAGE MODELS

## ABSTRACT

The human brain has long inspired the pursuit of artificial intelligence (AI). Recently, neuroimaging studies provide compelling evidence of alignment between the computational representation of artificial neural networks (ANNs) and the neural responses of the human brain to external stimuli, suggesting that ANNs may employ brain-like information processing strategies. While such alignment has been observed across sensory modalities—visual, auditory, and linguistic—much of the focus has been on the behaviors of artificial neurons (ANs) at the population level, leaving the functional organization of individual ANs that facilitates such brain-like processes largely unexplored. In this study, we bridge this gap by directly coupling sub-groups of artificial neurons with functional brain networks (FBNs), the foundational organizational structure of the human brain. Specifically, we extract representative patterns from temporal responses of ANs in large language models (LLMs), and use them as fixed regressors to construct voxel-wise encoding models to predict brain activity recorded by functional magnetic resonance imaging (fMRI). This framework effectively links the AN sub-groups to FBNs, enabling the delineation of brain-like functional organization within LLMs. Our findings reveal that LLMs (BERT and Llama 1–3) exhibit brain-like functional architecture, with sub-groups of artificial neurons mirroring the organizational patterns of well-established FBNs. Notably, the brain-like functional organization of LLMs evolves with the increased sophistication and capability, achieving an improved balance between the diversity of computational behaviors and the consistency of functional specializations. This research represents the first exploration of brain-like functional organization within LLMs, offering novel insights to inform the development of artificial general intelligence (AGI) with human brain principles.

## 1 INTRODUCTION

The human brain, with its unparalleled capacities in perception, cognition, reasoning, and creativity, stands as the pinnacle of biological intelligence and complexity (Sporns et al., 2000; Bassett & Gazzaniga, 2011). Understanding the mechanisms behind these cognitive abilities has been one of the most formidable challenges in neuroscience for decades (Brodmann, 1909; Hubel & Wiesel, 1979; Belliveau et al., 1991; Bear et al., 2020). Despite significant advances, the intricate processes through which the brain organizes and interprets information—transforming raw sensory inputs into meaningful representations that guide behavior and decision-making—remain elusive. Unraveling these complexities would not only deepen our understanding of human cognition but also lay the foundation for creating machines that truly emulate the intelligence of the human brain (Nilsson, 2009; Lu et al., 2018; Zhao et al., 2023b).

In recent years, artificial neural networks (ANNs), initially inspired by the architecture of the human brain, have emerged as one of the most promising technologies in quest to build machines capable of cognitive functions. Modern ANNs have groundbreaking abilities in fields like visual recognition (He et al., 2016; Dosovitskiy, 2020; Oquab et al., 2023), language processing (Vaswani, 2017; Devlin et al., 2018; Brown et al., 2020), and even reasoning tasks—domains that were once considered exclusive to human intelligence. These networks, with their remarkable capacity to learn complex patterns from vast amounts of data, have ignited hope that machines may one day replicate or even surpass human cognitive abilities. Yet, as we edge closer to this possibility, one fundamental

question lingers: Can artificial systems be powered by the same organizational principles that underpin human intelligence? This question becomes even more pertinent in light of recent advances in neuroimaging, which have revealed striking alignment between the computational representations of ANNs and the neural responses of the human brain. Through functional magnetic resonance imaging (fMRI), researchers have shown that ANN representations of external stimuli—whether visual (Zhao et al., 2023a; Yamins & DiCarlo, 2016; Kriegeskorte, 2015), auditory (Zhou et al., 2023; Li et al., 2023; Millet et al., 2022; Tuckute et al., 2023), or linguistic (Liu et al., 2023; Caucheteux & King, 2022; Schrimpf et al., 2021; Oota et al., 2024)—exhibit patterns that closely mirror patterns observed in the brain. This fascinating correspondence has led to a tantalizing hypothesis: that ANNs, in their computational architectures, might develop information processing strategies akin to those employed by the brain itself.

However, while these findings are compelling, most studies have focused on population-level comparisons between ANNs and brain activity. This offers a broad view of alignment but lacks the granularity required to uncover the functional organization within ANNs. In neuroscience, it is well established that the brain is composed of specialized regions, each responsible for distinct cognitive functions, which together form functional brain networks (FBNs) (Bassett & Bullmore, 2006; Power et al., 2011; Park & Friston, 2013). These networks are dedicated to processing specific types of information, such as visual input or linguistic content, and their interactions are essential for the brain's overall functionality (Smith et al., 2009). Similarly, it is plausible that within ANNs, a similar form of organization may exist, where sub-groups of artificial neurons play specialized roles in processing information.

Building on this inspiration, our study seeks to explore the functional organization of ANNs by directly coupling sub-groups of artificial neurons with FBNs. We focus on large language models (LLMs), such as BERT (Devlin et al., 2018) and the Llama family (Touvron et al., 2023a;b; Dubey et al., 2024), which are composed of vast numbers of artificial neurons capable of processing complex linguistic information. To achieve this, we extract representative temporal patterns from the activity of artificial neurons in LLMs and use them as fixed regressors in voxel-wise encoding models. These models enable us to predict brain activity recorded via fMRI, thus linking the sub-groups of artificial neurons in LLMs with their corresponding functional brain networks in the human brain. Our findings demonstrate that sub-groups of artificial neurons in LLMs align closely with the functional interactions within well-established brain networks. By analyzing the evolution of these relationships across four LLMs (BERT and Llama 1-3), we observe that the brain-like functional organization in these models becomes increasingly pronounced as their capabilities grow, achieving an improved balance between the diversity of computational behaviors and the consistency of functional specializations. This research is the first to characterize the brain-like functional organization within LLMs, providing keys insights that could shape the future development of brain-inspired AI systems and engineer brain-like intelligence.

## 2 RELATED WORK

### 2.1 NEURAL ENCODING OF COMPUTATIONAL LANGUAGE MODELS

Neural encoding studies have demonstrated that computational language models based on deep neural networks exhibit considerable representational alignment to neural activity in the human brain (Abdou, 2022; Schrimpf et al., 2021; Oota et al., 2024; Antonello et al., 2024). Most prior research has adopted linear encoding models to map between the computational representation of language models and neural responses elicited by the same set of stimuli. Although these studies have yielded promising results, they primarily focus on modeling the behaviors of ANs at the population level. Specifically, they utilize layer-level embeddings—comprising a collection of individual AN's responses—to predict neural activity, thereby leaving the functional organization of individual ANs largely unexplored.

### 2.2 INTERPRETING BEHAVIORS OF INDIVIDUAL ANS

Researchers have developed various strategies to interpret the behaviors of individual ANs in computational language models (Zhao et al., 2024). These strategies include feature attribution, probing, neuron activation analysis, attention visualization, adversarial example, and inverse recognition,

among others (Wu et al., 2023; Zhang et al., 2022; Yeh et al., 2023; Wang et al., 2022). Recently, researchers have employed more advanced models such as GPT-4 to automate the interpretation of large scale individual ANs in less capable models such as GPT-2 (Bills et al., 2023). Singh et al. (2023) similarly use LLMs to generate candidate explanations for text modules, such as a neuron in LLM, based on the n-grams that elicit the most activation from the neuron. Synthetic data is then generated based on these explanations, and the neuron's activation to the data is assessed to identify the top candidate explanations. While these strategies provide valuable insights into our understanding of language models, the functional organization of individual ANs has rarely been explored. Furthermore, the behaviors of individual ANs have yet to be linked to neural response, leaving the question of whether the organization of ANs mirrors the functional structure and organization found in the brain inadequately addressed.

## 3 METHODS

### 3.1 OVERVIEW

The study overview is illustrated in Figure 1. We begin by defining artificial neurons (ANs) in LLMs and quantifying their temporal responses to external stimuli $\mathbf{X} \in \mathbb{R}^{t \times n}$ (Figure 1a). Subsequently, we employ a sparse representation (Mairal et al., 2009) scheme to learn a set of representative temporal response patterns, referred to as a dictionary $\mathbf{D}_{AN} \in \mathbb{R}^{t \times k}$ (Figure 1b). Afterwards, we use the dictionary $\mathbf{D}_{AN}$ as regressors to build voxel-wise encoding models to predict fMRI brain activity. The encoding coefficients associated with each atom reveal how that atom couples with functional activity of the entire brain (Figure 1c). By integrating this coupling relationship with the association between ANs and $\mathbf{D}_{AN}$ established during learning of representative temporal responses, we infer brain-like functional organization in LLMs.

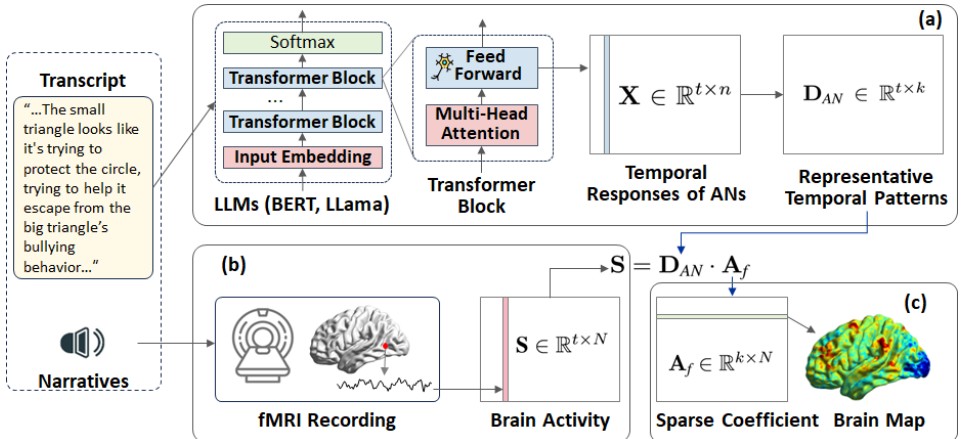

Figure 1: The study overview. We learn representative patterns $\mathbf{D}_{AN}$ (b) from the temporal responses of ANs in LLMs (a) and use $\mathbf{D}_{AN}$ as regressors to reconstruct fMRI brain activity recorded by fMRI (c). Atoms in $\mathbf{D}_{AN}$ selectively activate specific brain areas/networks.

### 3.2 ARTIFICIAL NEURONS IN LLMS AND THEIR TEMPORAL RESPONSES

In this study, we focus on four LLMs: the pre-trained BERT model (Devlin et al., 2018), which serves as a foundational transformer-based language model, and three progressively advanced models from the evolutionary Llama family, Llama 1-3 (Touvron et al., 2023a;b; Dubey et al., 2024). BERT, with its bidirectional encoder, is a widely recognized baseline model to capture rich, contextualized word representations. In contrast, the Llama models, employing a decoder-based architecture, represent a more advanced, contemporary approach, exhibiting superior performance across diverse tasks. Examining the evolution of these LLMs may offer insights into the development of functional organization within these models.

Building on the established definitions of Artificial Neurons (ANs) in large language models (LLMs) (Bills et al., 2023; Samek et al., 2021), we define each neuron in the second fully connected layer of the feed-forward network within each transformer block as an individual AN (Figure 1a). In BERT, this applies to the encoder blocks, while in the Llama models, it applies to the decoder blocks. With this definition, the number of ANs corresponds to the dimensionality of the output embedding in each transformer block. For instance, BERT consists of 12 layers, yielding 9,216 ANs (12 layers × 768 dimensions per layer), whereas the Llama models, with 32 layers, define 131,072 ANs (32 layers × 4096 dimensions per layer).

Given a text input, the temporal responses of each AN are formally defined as it activations in response to the sequence of input tokens. The temporal responses of all ANs at layer $l$ can be readily obtained through the layer's output $\mathbf{X}_l \in \mathbb{R}^{t \times n_l}$, where $t$ is the the number of tokens in the input sequence, and $n$ is the number of ANs at layer $l$, corresponding to the dimensionality of the output.

Additionally, it is critical to synchronize the temporal responses of artificial neurons (ANs) with the fMRI timeline. To achieve this, we align the text tokens with the corresponding fMRI volumes using the time-stamped word-level transcripts from the Narratives fMRI dataset (Nastase et al., 2021). However an fMRI volume generally spans multiple text tokens, we follow common practice in brain encoding studies by averaging the ANs' responses over these tokens within each fMRI time interval. This produces a temporal response curve that matches the length of the fMRI sequence. Finally, we convolve the temporal response curve of each AN with a canonical hemodynamic response function (HRF) implemented in SPM[1], to account for the hemodynamic delay inherent in fMRI recordings.

### 3.3 Representative Temporal Response Patterns of ANs

Identifying representative temporal response patterns of ANs is crucial for simplifying the analysis given the vast number of ANs in models like BERT and Llama. With thousands of ANs in each model (e.g., 9,216 in BERT and 131,072 in Llama), analyzing individual responses is not only impractical but also risks obscuring key trends due to factors such as noise and self-correlation among the ANs. In this study, we employ a sparse representation scheme (Mairal et al., 2009) to learn a set of representative patterns from the temporal responses of the entire group of ANs.

Given the set of temporal responses $\mathbf{X} \in \mathbb{R}^{t \times n}$, where $n$ is the total number of ANs and $t$ is the length of the temporal responses, the objective is to find a sparse representation $\mathbf{A}_{AN} \in \mathbb{R}^{k \times n}$ over a dictionary $\mathbf{D}_{AN} \in \mathbb{R}^{t \times k}$, minimizing the reconstruction error while imposing a sparsity constraint on $\mathbf{A}_{AN}$ (Mairal et al., 2009):

$$\min_{\mathbf{A}_{AN}} \|\mathbf{X} - \mathbf{D}_{AN}\mathbf{A}_{AN}\|_2 + \lambda_{AN} \|\mathbf{A}_{AN}\|_1 \tag{1}$$

where $\lambda_{AN}$ is a regularization parameter that controls the trade-off between reconstruction accuracy and sparsity of $\mathbf{A}_{AN}$. In this context, $\mathbf{D}_{AN}$ represents a set of basis vectors or atoms, which is the representative temporal patterns that capture the essential dynamics of the ANs' temporal responses. The sparsity constraint ensures that each temporal response is characterized by only a few key patterns. By learning this dictionary, we can express the entire set of temporal responses $\mathbf{X}$ as a combination of these representative patterns, weighted by the sparse coefficients in $\mathbf{A}_{AN}$.

### 3.4 Voxel-wise Encoding of fMRI Brain Activity

We construct voxel-wise encoding models to establish the relationship between ANs and brain activities. This approach allows us to determine how the temporal responses of ANs can predict or account for the neural signals captured by fMRI. The encoding models are based on a similar scheme to the one used for learning representative temporal response pattern. The key difference is that we fix the dictionary $\mathbf{D}_{AN}$ to learn a sparse representation $\mathbf{A}_f \in \mathbb{R}^{k \times N}$ for reconstructing the fMRI brain activity $\mathbf{S} \in \mathbb{R}^{t \times N}$, where $N$ is the number of voxels:

$$\min_{\mathbf{A}_f} \|\mathbf{S} - \mathbf{D}_{AN}\mathbf{A}_f\|_2 + \lambda_f \|\mathbf{A}_f\|_1 \tag{2}$$

---

[1]https://www.fil.ion.ucl.ac.uk/spm/

Each row in $\mathbf{A}_f$ indicates the importance of the corresponding atom of $\mathbf{D}_{AN}$ in reconstructing fMRI brain activities at each voxel (Figure 1c). It is noted that the voxel-wise encoding models are constructed for each subject independently. A one-sample $t$-test over the entire population of subjects is conducted for each voxel to examine whether the encoding coefficient of a given atom $\mathbf{D}_{AN}$ is above chance level (FDR corrected). Rather than simply showing voxel-level activations, this statistical map provides a spatial depiction of the brain regions linked to each representative pattern, offering a more interpretable perspective. For simplification, we refer to this as a brain map.

### 3.5 RELATIONSHIP INFERENCE BETWEEN ANs AND BRAIN NETWORKS

The representative temporal response patterns $\mathbf{D}_{AN}$ serve as a bridge between ANs and brain activity. Specifically, the $i^{th}$ AN can be associated with the $j^{th}$ atom of $\mathbf{D}_{AN}$ where $\mathbf{A}_{AN}(\cdot, i)$ is maximized. In this way, each atom in $\mathbf{D}_{AN}$ corresponds to a subset of ANs. Simultaneously, the voxel-wise encoding models link each atom in $\mathbf{D}_{AN}$ to specific brain regions, forming a brain map that typically spans multiple brain networks, such as auditory, language and visual networks. We utilize a network correspondence tool (Kong et al., 2024) to automatically identify the brain networks involved in these brain maps by referring to the 17 FBNs reported previously (Yeo et al., 2011). This approach allows us to infer the relationship between subsets of ANs and brain networks, revealing how these subsets and corresponding temporal patterns align with the brain's functional architecture.

### 3.6 IMPLEMENTATION DETAILS

We use the "Narratives" fMRI dataset (Nastase et al., 2021) in this study. The fMRI data were acquired when human subjects listened to 27 spoken stories and released with various pre-processed versions. We use the AFNI-nosmooth version of one fMRI session, the "Shapes", due to the high spatial resolution ($2 \times 2 \times 2\text{mm}^3$), adequate number of subjects (59 subjects) and the integrity of the narrative stimuli. The fMRI volumes before the onset and after the end of the story are discarded. The time courses of each voxel is normalized to have unit norm. For the LLMs, we use the pre-trained BERT [2] and Llama family [3] (Llama1-7B, Llama2-7B and Llama3-8B). In the sparse representation of ANs' temporal responses, the dictionary size ($k$) is set as 64, and the sparsity constraint parameter $\lambda_{AN}$ is set as 0.15 for all the LLMs. The $\lambda_f$=0.08 is used in the sparse reconstruction of fMRI activity.

## 4 RESULTS

### 4.1 SPARSE REPRESENTATION OF ANs AND BRAIN ACTIVITY

The temporal responses of ANs can be effectively represented by the dictionary $\mathbf{D}_{AN}$, as evidenced by the high $R^2$ values shown in Figure 2(a). Among the Llama family, the $R^2s$ values are comparable, with measurements of 0.5021±0.1119, 0.5005±0.1114 and 0.5032±0.1139, respectively. The BERT model demonstrates relatively higher $R^2$ values (0.6005±0.1127) compared to the Llama family. This discrepancy may be attributed to the significantly smaller number of ANs in BERT compared to the Llama models, which results in lower diversity of temporal response patterns of the ANs and consequently improves the performance of dictionary learning.

The distribution of the $R^2s$ vaules for the sparse reconstruction of fMRI brain activity across the four LLMs shown in Figure 2(b) indicates that the Llama family correlates more closely with brain activity compared to BERT, which is in accordance with their performance in natural language tasks. Meanwhile, the spatial distributions of the $R^2s$ across the four LLMs (Figure 2c-f) show considerable alignment with one another. The brain regions with superior encoding performance encompass the superior and middle temporal lobes, lateral and medial occipital lobes, angular gyrus, anterior and posterior cingulate cortices, temporo-parietal junction, and wide spread areas in the frontal lobe. This spatial distribution pattern, as well as the range of $R^2$, closely resembles the brain scores from previous studies on neural encoding of natural language processing models (Antonello et al., 2021; Schrimpf et al., 2021; Caucheteux & King, 2021), validating the reliability of the voxel-wise encoding models in this study.

---

[2]https://huggingface.co/docs/transformers/model-doc/bert
[3]https://huggingface.co/meta-llama/

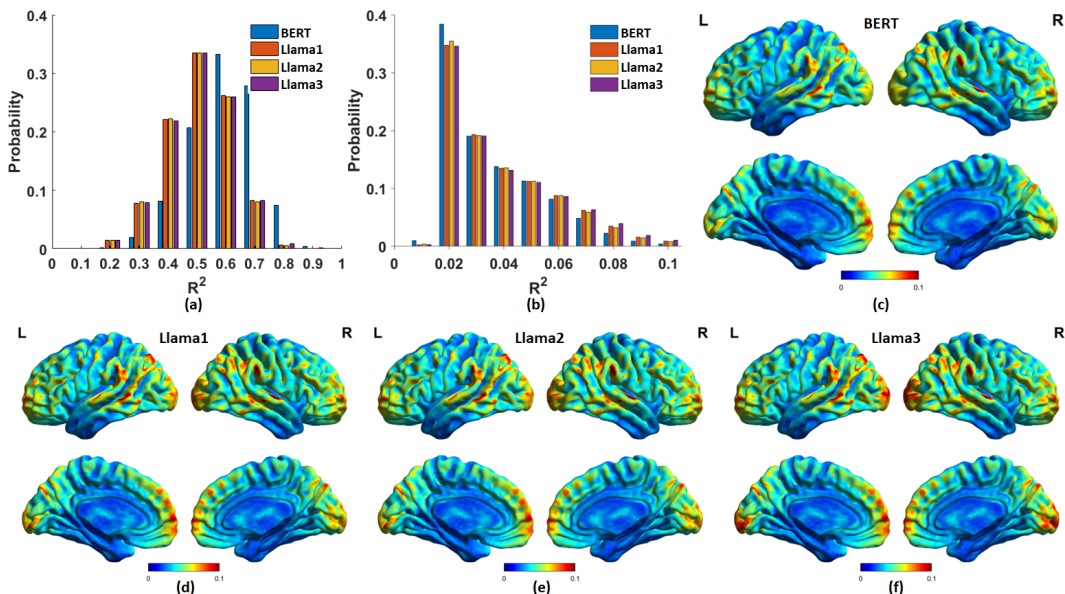

Figure 2: The $R^2$ values in the sparse representation of temporal responses of ANs (a) and in the sparse reconstruction of fMRI activity (b) using $\mathbf{D}_{AN}$. (c-f): The spatial distribution of the $R^2s$ in BERT and Llama family visualized on the cortical surface, respectively.

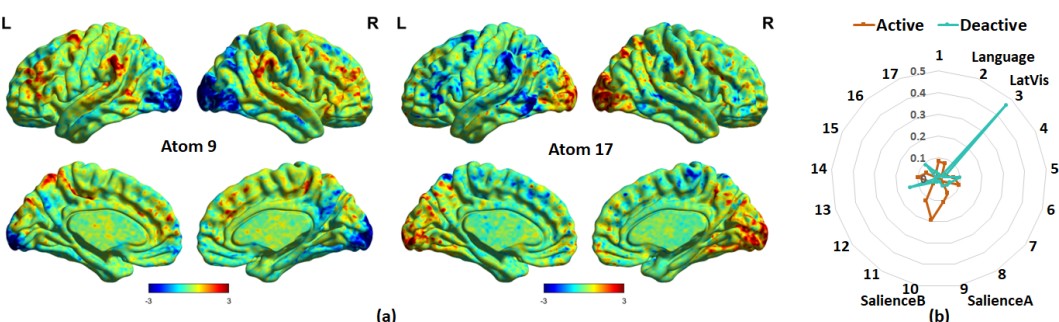

Figure 3: (a) Two exemplar brain maps corresponding to the atom #9 and atom #17 in Llama3. (b) Automatic brain network labelling of atom #9, illustrating the activation of the language, salienceA and salienceB networks, and the deactivation of the lateral visual (LatVis) cortex.

## 4.2 BRAIN MAP ANALYSIS

The brain maps reveal intricate functional interactions and competitions among well-established FBNs. Figure 3 shows two exemplar brain maps corresponding to atom 9 and atom 17 in Llama3 (Figure 3a), along with the automatic brain network labelling of atom 9 which exhibits the concurrent activation of the language, salienceA and salienceB networks, and the deactivation of the lateral visual (LatVis) cortex (Figure 3b). A comprehensive visualization for all the 64 brain maps across the four LLMs is provided in A.1.

We observed notable variability in the involvement of FBNs in brain maps across different FBNs (Figure 4a). A subset of FBNs, including the LatVis cortex, language network, default mode network (DMN), working memory (WM) network, primary auditory cortex, salience network, fronto-parietal network (FPN) and dorsal attention network (DAN), are more frequently engaged in brain maps, with both positive (activation) and negative (deactivation) involvement. On the contrary, the meidal visual (MedVis) cortex, parietal memory (ParMemory) cortex, sensorymotor network (SMN), LimbicA and LimbicB are less frequently involved. Notably, the patterns of FBN engagement in brain maps are consistent across the four models.

In line with previous findings on neural language processing, our results highlight the engagement of functionally specialized brain regions/networks including the primary auditory cortex, visual cortex, language and FPN (Friederici, 2011; Caucheteux & King, 2022; Schrimpf et al., 2021). More importantly, our results further underscore the importance of domain-general brain regions/networks in this process, particularly the DMN, WM network and DAN. These findings are consistent with previous neuroimaging studies using dynamic naturalistic stimuli (e.g., auditory stories and movies), which suggest that the DMN plays a key role in integrating incoming extrinsic information, temporarily stored in the WM, with prior intrinsic information over relatively long timescales to form context dependent models (Yeshurun et al., 2021).

The brain maps also exhibit relatively complex functional interactions among FBNs. Specifically, most brain maps involve the concurrent activation or deactivation of multiple FBNs, as illustrated by the distribution of the brain maps associated with different number of FNBs (Figure 4b). In this context, our experimental results highlight the cooperative interaction of FBNs in neural language processing (Horwitz & Braun, 2004; Schoffelen et al., 2017; Fedorenko et al., 2024).

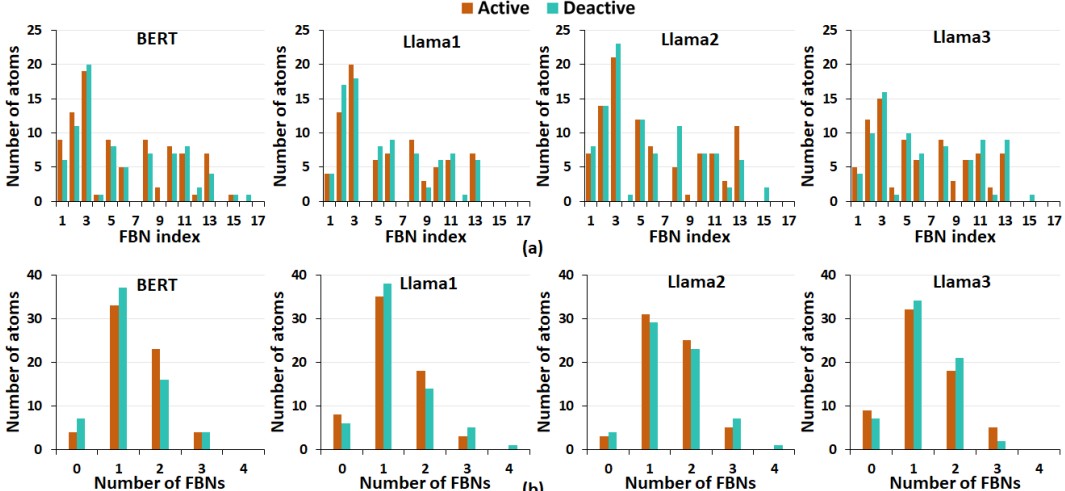

1: Auditory  2: Language  3: LatVis (lateral visual)  4: MedVis (medial visual)  5: DMN-A (default mode network A)  6: DMN-B  7: DMN-C  8: WM (working memory)  9: SalienceA  10: SalienceB  11: FPN (fronto-parietal network)  12: ParMemory (parietal memory)  13: DAN-A (dorsal attention network A)  14: DAN-B  15: SMN (sensorimotor network)  16: LimbicA  17: LimbicB

Figure 4: (a) The number of brain maps associated with different FBNs. (b) The distribution of the brain maps associated with different number of FBNs.

## 4.3 Evolution of Brain-like Functional Organization within LLMs

We present a detailed analysis of the FBN components involved in brain maps across the four models in Figure 5, with the aim of exploring the evolution of brain-like organization patterns within these models. The color-coding in Figure 5 represents the Dice coefficient obtained from FBN labelling (Kong et al., 2024), quantifying the spatial overlap between brain maps and FBNs. Positive and negative values denote activation and deactivation of FBNs, respectively. The y-axis is the FBN index, reordered in descending order according to the frequency of FBN involvement in brain maps (both activation and deactivation), with the actual reordered indices provided at the bottom of each sub-figure. The x-axis is the brain map index, organized according to the presence order of FBNs (activation first, followed by deactivation). One noteworthy observation is that a greater number of brain maps with identical FBN labels appears in more advanced LLMs, as highlighted by the braces in Figure 5. In addition, the overall distribution of FBN involvement is noticeably sparser in Llama3 compared to other models. This observation suggests that more advanced LLMs may promote more compact brain-like functional organizations. One possible explanation for this observation is that more advanced LLMs tend to learn more compact representational policies and integrate these policies more efficiently to achieve improved performance on language tasks.

It is hypothesized that brain maps with identical FBN labels share similar functional interactions among the associated FBNs. To test this hypothesis, we focused on a subset of brain maps displaying

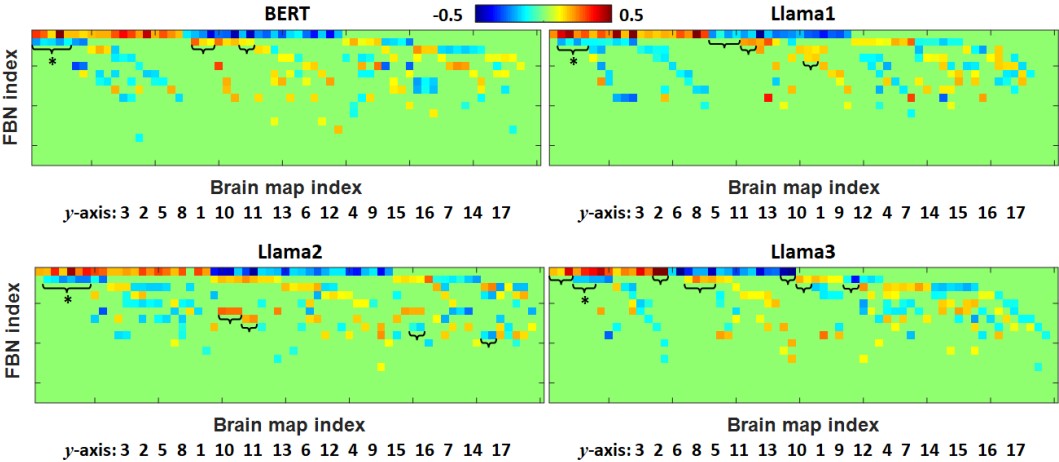

Figure 5: The detailed FBN components involved in brain maps. The color-coding represents the Dice coefficients obtained from FBN labeling, which measures the spatial overlap between brain maps and FBNs. A negative Dice value here indicates deactivation of FBNs in a given brain map. Braces are used to highlight brain maps that have identical FBN labels.

functional interactions between the LatVis (activation) and the language network (deactivation), a pattern consistently observed across the four LLMs (Figure 5, braces with stars). For each LLMs (Figure 6a-d), we show one exemplar brain map (Figure 6, first column) from this subset (subset size: 5/4/6/3 for BERT/Llama1/Llama2/Llama3, respectively), and evaluate the temporal consistency of the subset by calculating the inter-atom Pearson correlation coefficients (Figure 6, second column) of their temporal responses (columns in $\mathbf{D}_{AN}$). We also illustrate the distribution of number of ANs on LLM layers (Figure 6, third column).

Our results show that the variability of temporal correlation coefficients decrease sequentially in BERT, Llama1, Llama2 and Llama3, as evidenced by the standard deviations (0.2141, 0.1765, 0.1603 and 0.0233 for the four models, respectively). The high variability in BERT, Llama1 and Llama2 indicates that the subset of atoms in these models exhibit distinct functional processing patterns, despite involving identical FBNs. Meanwhile, the subset of atoms in Llama3 shows the highest temporal consistency (0.236, Figure 6e) compared to other models. Notably, the moderate value of temporal consistency in Llama3 implies a coexistence of both shared and distinctive functional processing patterns among those atoms. These findings provide novel evidence for the principle of functional organization in LLMs: the ANs in more advanced models are organized to achieve an enhanced balance between the diversity of computational behaviors and the consistency of functional specializations.

To further investigate the properties of those atoms within LLMs, we identified the ANs that anchor to a each specific atom and evaluated the consistency of their distribution pattern on LLM layers by calculating the average Pearson correlation coefficient over all possible atom pairs. Our results (Figure 6f) show that the AN distribution patterns are more consistent in Llama3 compared to BERT, Llama1 and Llama2, suggesting a more hierarchical organization of ANs within Llama3. Intriguingly, we observed a greater concentration of ANs in the deeper layers of Llama3. Given that this subset of atoms reveals the activation of LatVis and deactivation of the language network, this finding resonates with neuroscience evidence suggesting that visual imagery is represented at a higher level of the language hierarchy (Speed et al., 2024; Zwaan, 2003; Bergen et al., 2007), highlighting the potential for Llama3 to capture complex linguistic and cognitive processes.

## 5 CONCLUSION AND DISCUSSION

In this study, we explored the brain-like functional organization within LLMs. We built a neural encoding model that uses the representative patterns learned from the temporal responses of AN populations defined in LLMs as fixed regressors to predict functional brain activity. These representative patterns serves as a bridge between AN sub-groups to functional brain networks, enabling

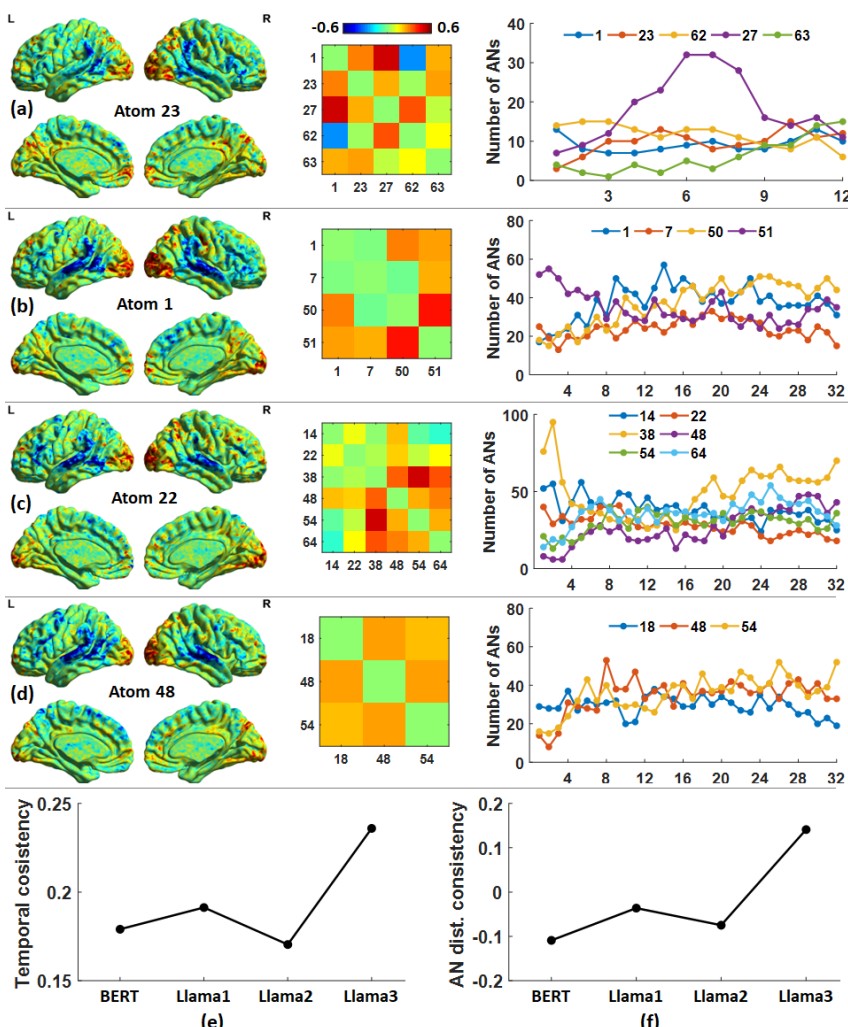

Figure 6: The subset of atoms that reveals functional interaction between LatVis (activation) and the language network (deactivation) across the four LLMs. (a-d) are for BERT, Llama1, Llama2 and Llama3, respectively, showing the brain maps (column 1), Pearson correlation coefficients between the temporal responses of atom-pairs (column 2), and the distribution pattern of ANs on LLM layers (column 3, the *x*-axis is layer index). (e) The temporal consistency of atoms. (f) The consistency of the distribution patterns of ANs on LLM layers.

us to disentangle how individual ANs within LLMs are functionally organized to support their unprecedented capabilities in language tasks. The proposed framework addresses a key limitations in previous research that examined the behaviors of artificial neurons at a population level, which has hindered a clear understanding of the functional organization within LLMs. Our experimental results demonstrate that the brain-like functional organization within LLMs evolves with their capabilities, where more advanced LLMs achieve an improved balance between diverse computational behaviors and consistent functional specializations.

The present study acknowledges several limitations. First, we fixed the number of atoms (dictionary size) in the dictionary which describes the representative patterns of temporal responses of ANs, despite the fact that the number of ANs varies across different LLMs. Identifying model-specific dictionary size may facilitate a more accurate depiction of the brain-like functional organization within LLMs. Second, our assessment of the coupling relationships between AN sub-groups and functional brain networks was conducted for only a limited number of atoms. However, these coupling relationships described by the remaining atoms could carry valuable clues to investigate neural language

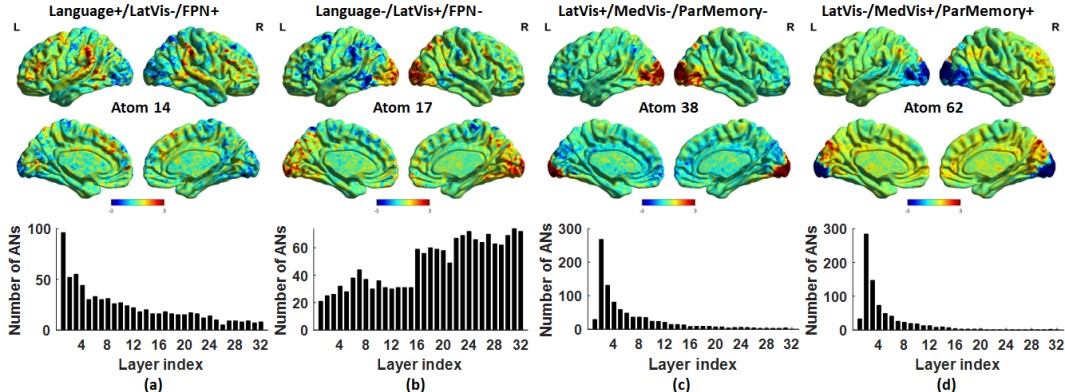

Figure 7: (a-b) Two atoms in Llama3 with opposite brain activity patterns and distributions of ANs on layers. (c-d) Two atoms with opposite brain activity patterns but similar distributions of ANs on layers. "+" and "-" represent activation and deactivation, respectively.

processing in the human brain. For example, Figure 7(a-b) illustrate two atoms in Llama3 exhibiting opposite brain activity patterns in the language network, FPN, and LatVis. In accordance, the corresponding distributions of ANs on layers display inverse patterns. On the contrary, Figure 7(c-d) show two atoms demonstrating opposite brain activity patterns in LatVis, MedVis and ParMemory, while the corresponding distributions of ANs on layers remain similar. Thus, future research could aim to further elucidate the brain-like functional organization within LLMs and the neural mechanism underlying language processing by linking brain activity patterns with ANs' computational behaviors, specifically their selective responses to external stimuli. Third, our experiments were limited to one fMRI session, validating and evaluating this framework on a larger scale fMRI cohort is essential for future studies. Finally, applying the proposed framework to the foundation models in other modalities could provide additional evidence regarding the brain-like functional organization in modern artificial general intelligence models.

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

## A APPENDIX

### A.1 THE BRAIN MAPS AND DISTRIBUTION OF ANS ON LAYERS

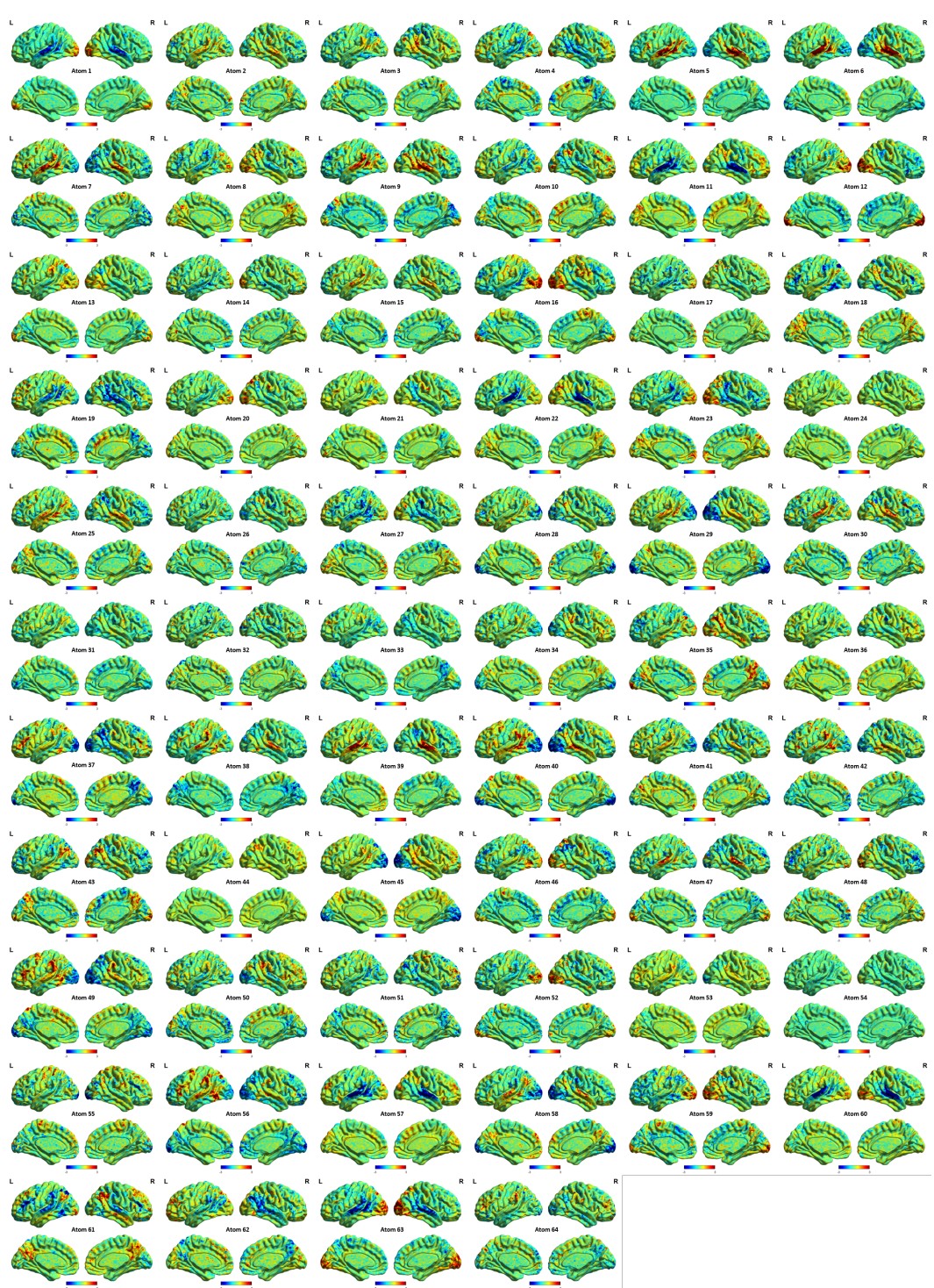

Figure 8: The visualization of brain maps for all the 64 atoms in BERT.

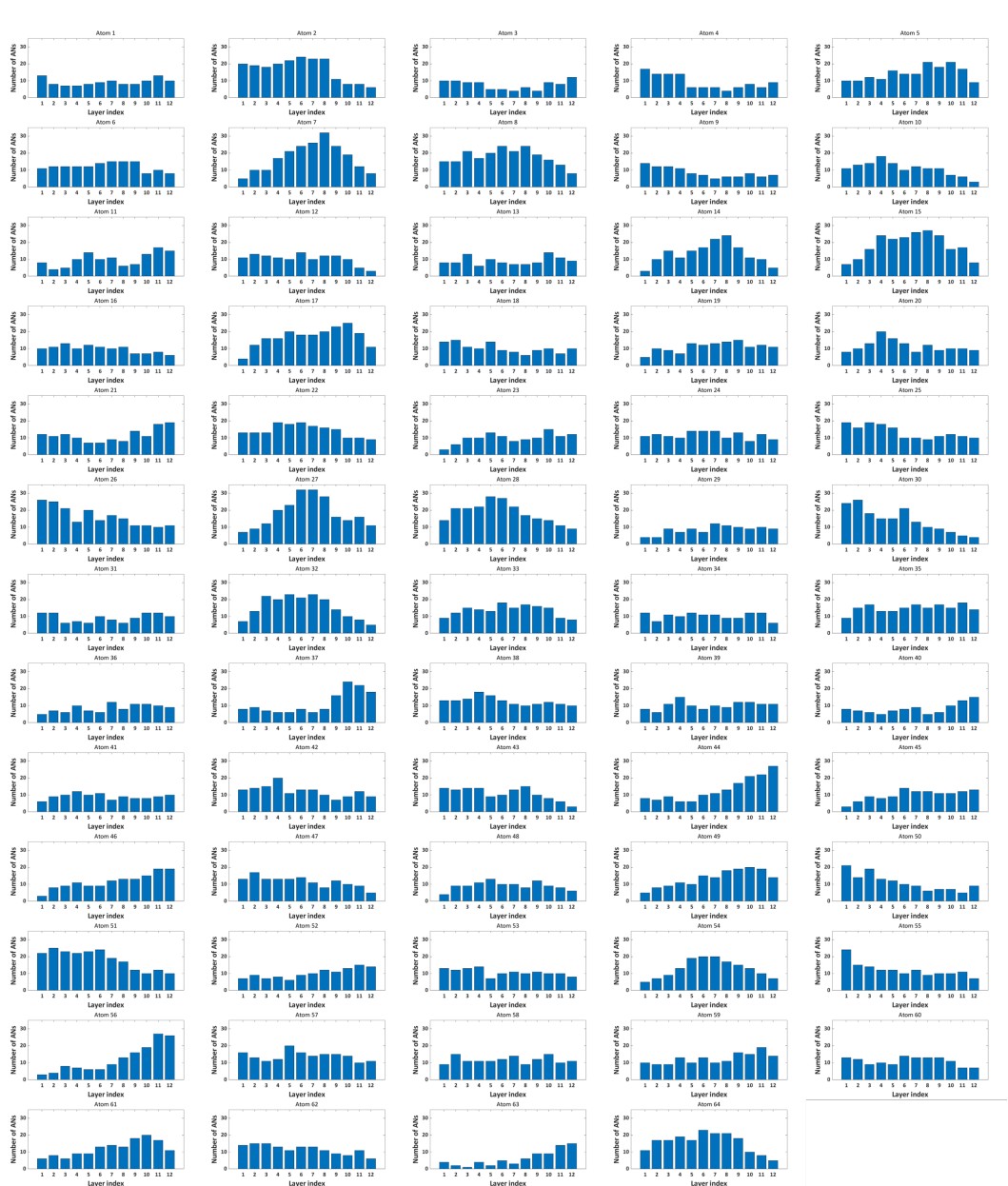

Figure 9: The distribution of ANs on layers for all the 64 atoms in BERT.

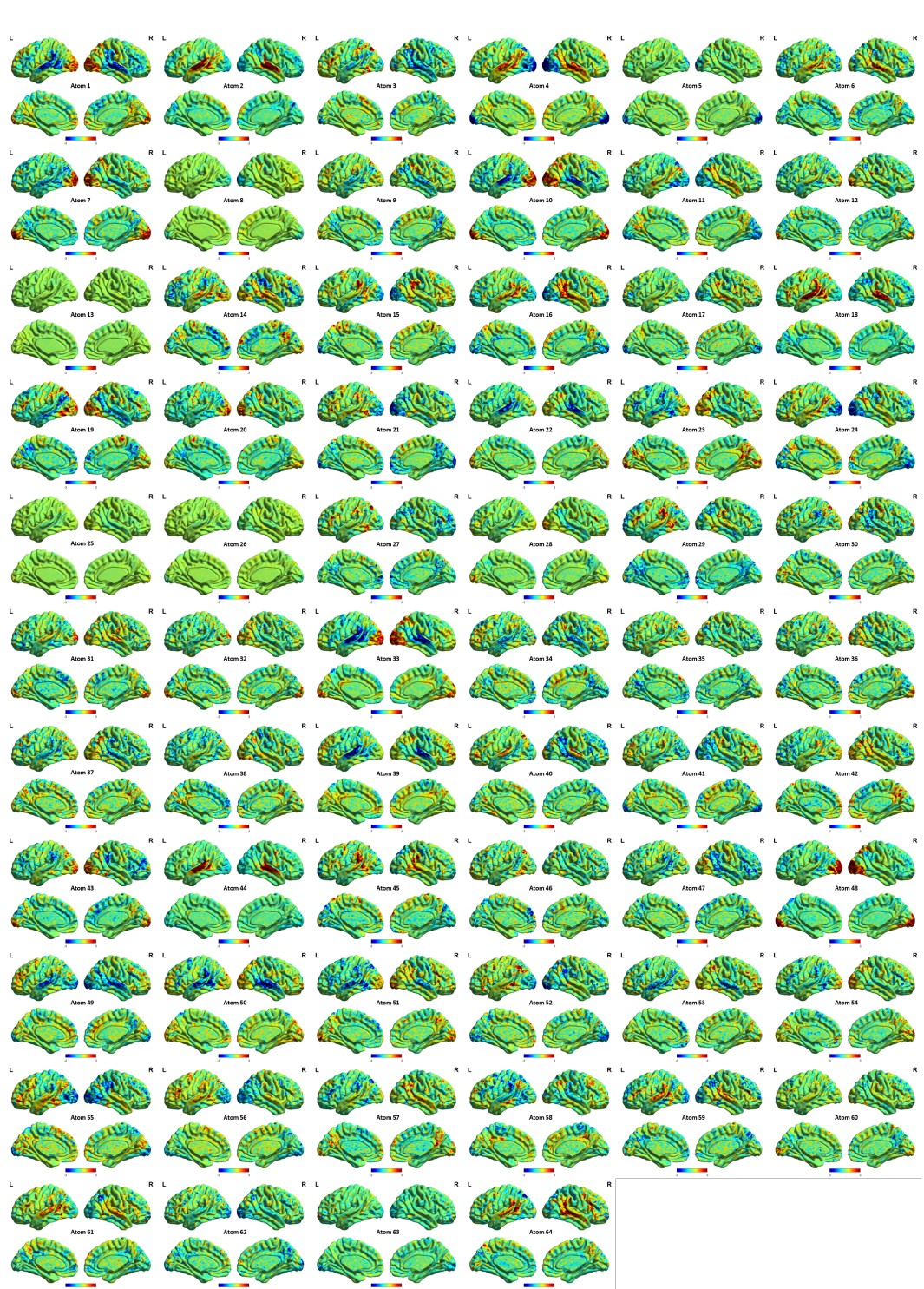

Figure 10: The visualization of brain maps for all the 64 atoms in Llama1.

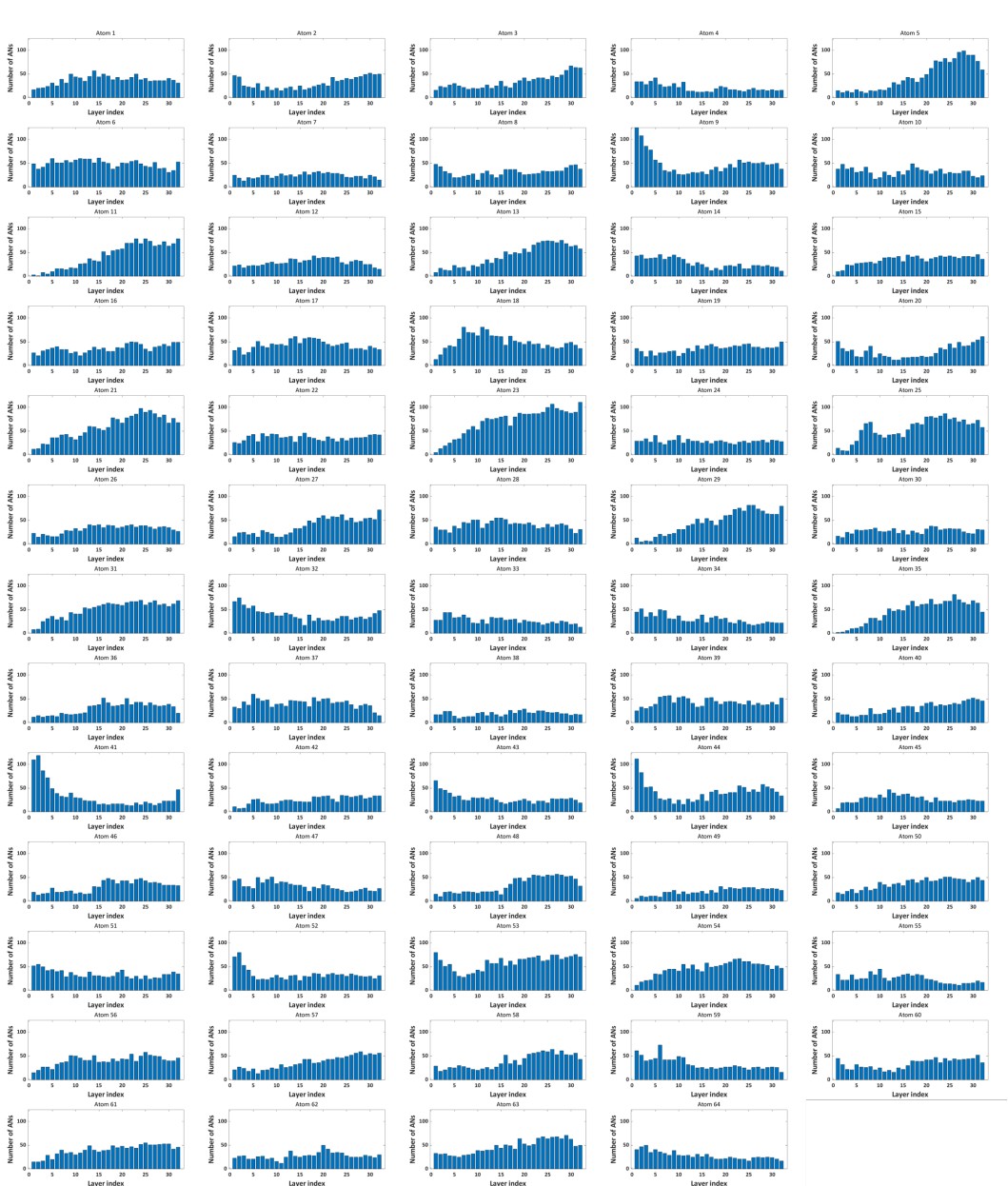

Figure 11: The distribution of ANs on layers for all the 64 atoms in Llama1.

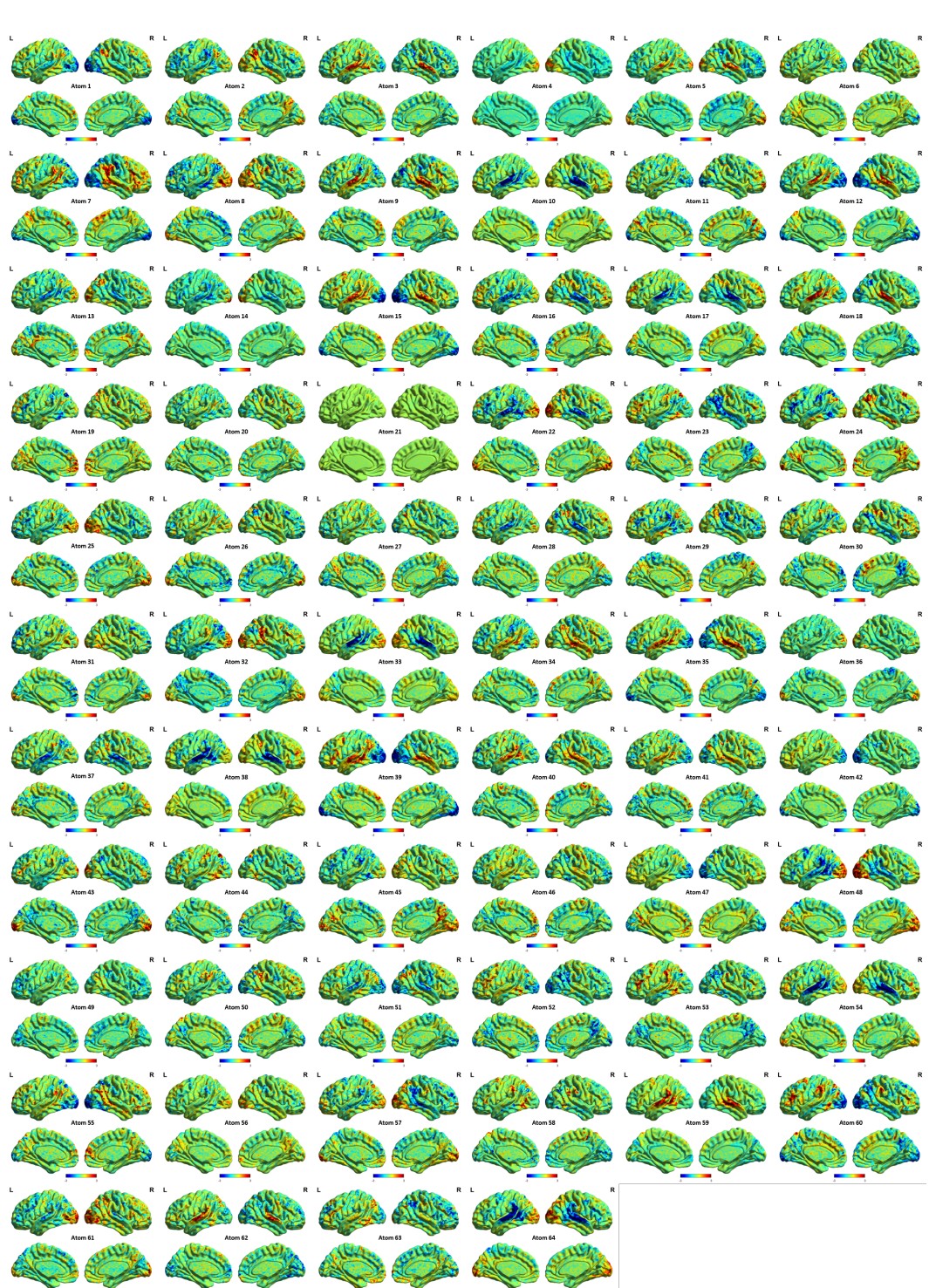

Figure 12: The visualization of brain maps for all the 64 atoms in Llama2.

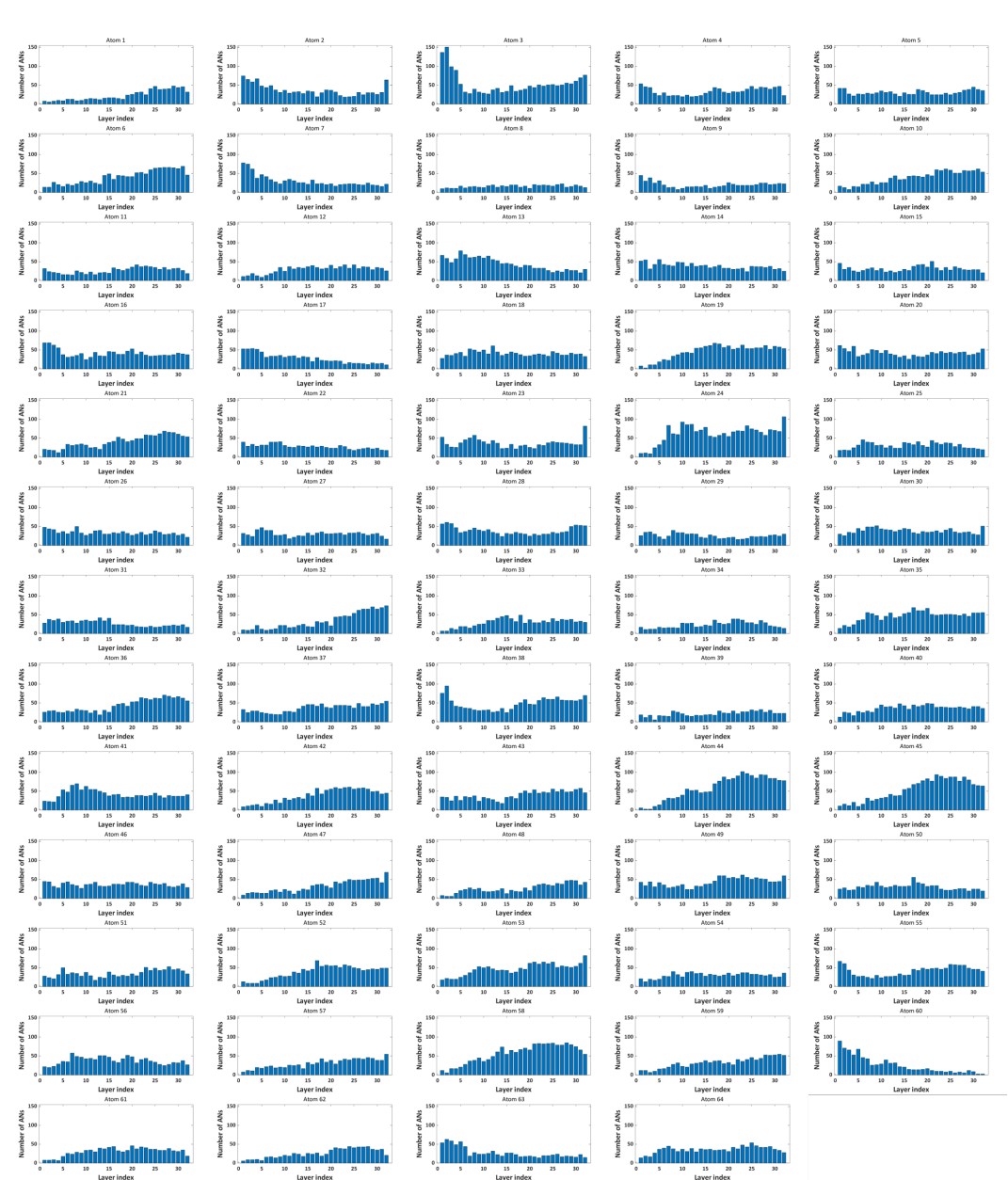

Figure 13: The distribution of ANs on layers for all the 64 atoms in Llama2.

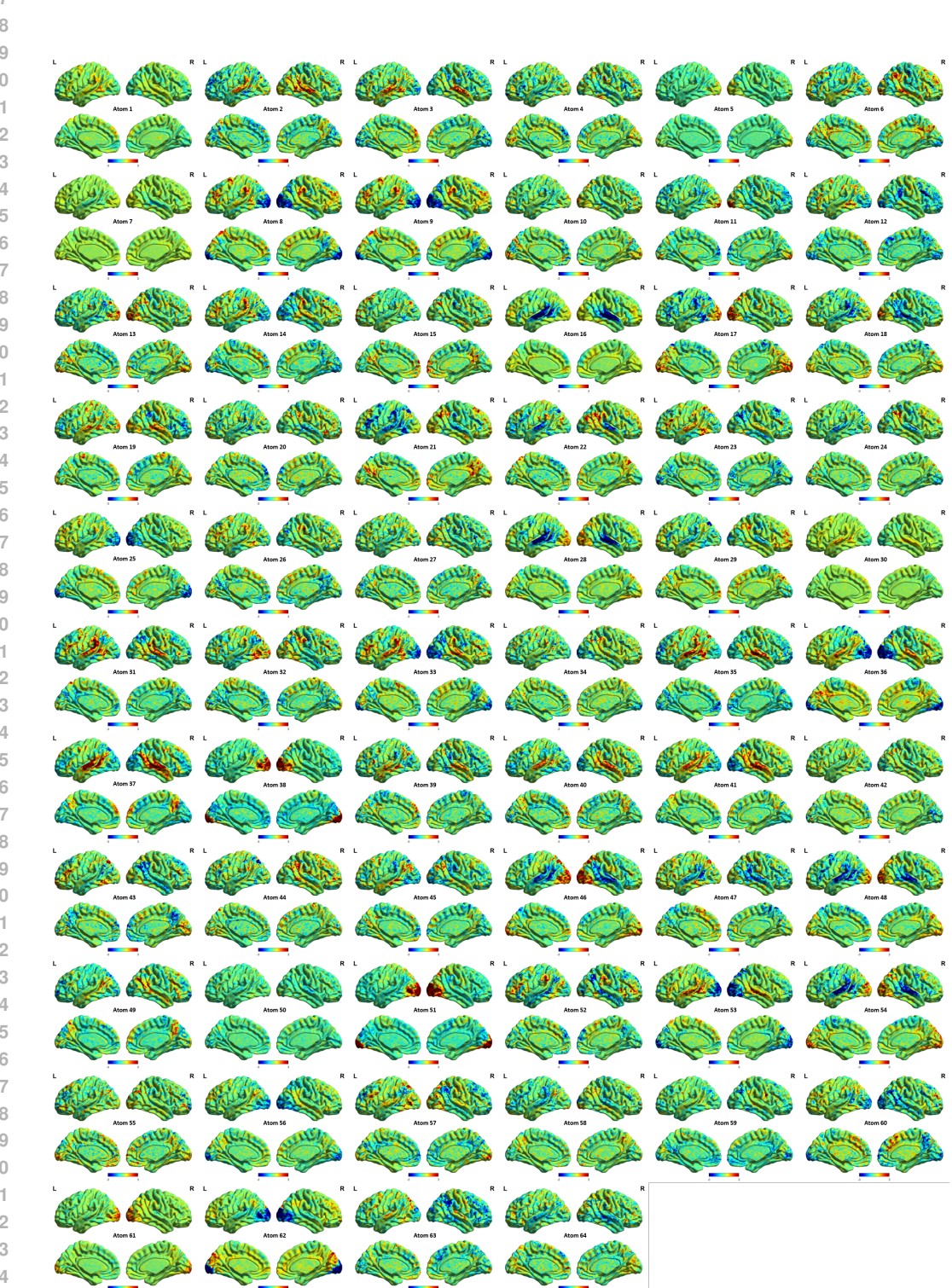

Figure 14: The visualization of brain maps for all the 64 atoms in Llama3.

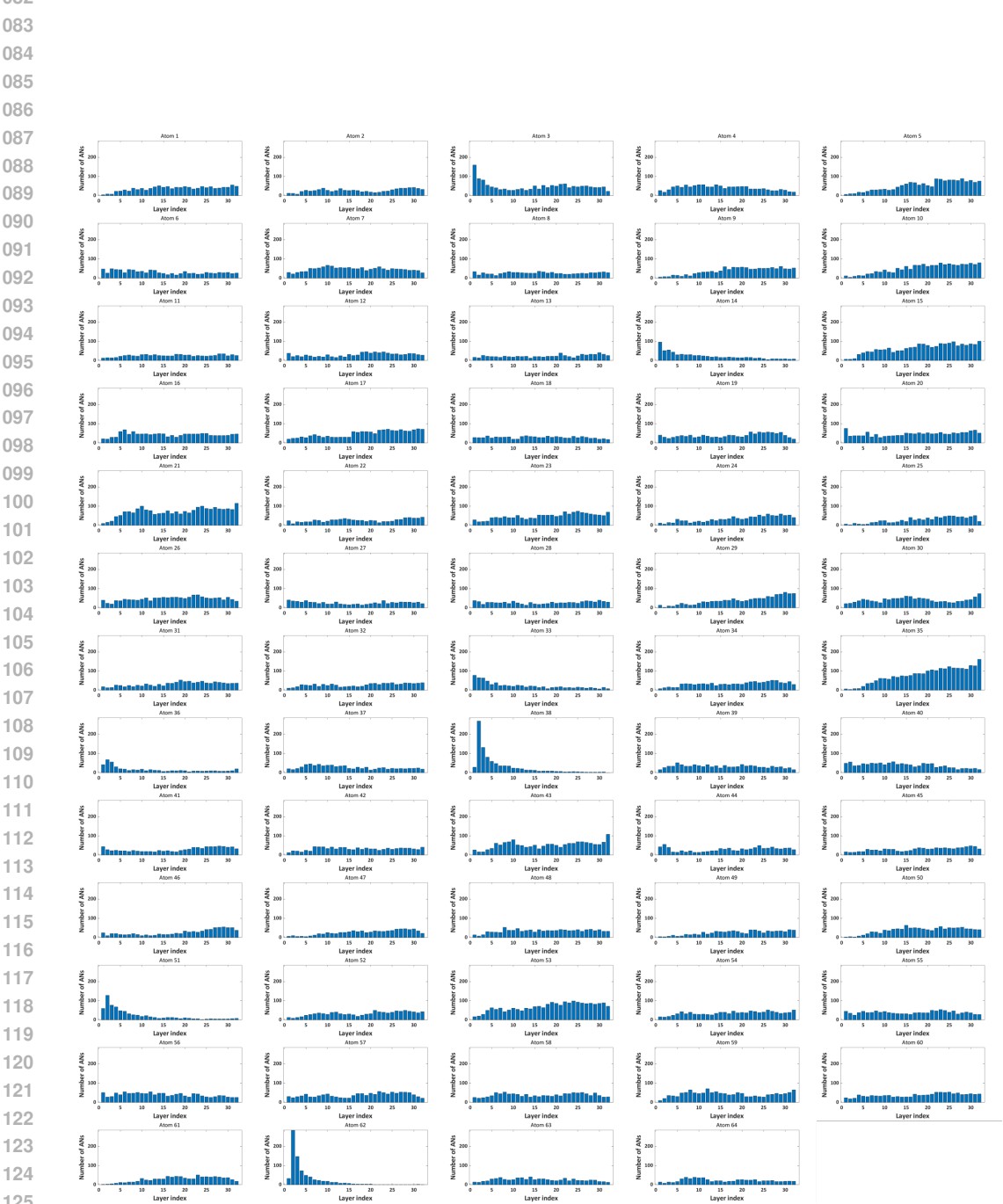

Figure 15: The distribution of ANs on layers for all the 64 atoms in Llama3.

