# OpenReview forum: "Brain-like Functional Organization within Large Language Models"
_ICLR.cc/2025/Conference — ICLR 2025 Conference Withdrawn Submission_

### Official Review · Reviewer_6fxY · 2024-10-29

**Soundness:** 3
**Presentation:** 1
**Contribution:** 1
**Rating:** 3
**Confidence:** 4

**Summary:**

In this paper, the authors relate the units of LLMs to different functional brain networks during narrative comprehension using functional magnetic resonance imaging (fMRI). First, the authors identify the temporal response pattern of artificial neurons in the LLM, and then use these response profiles as regressors to then map onto fMRI brain activity. They perform this on several common LLMs, including BERT and the Llama family of models. The authors’ main finding is that LLM activity maps onto meaningful clusters of brain activity (i.e., functional brain network patterns), suggesting that LLMs are organized similarly to the human brain.

**Strengths:**

The technical components and the methods in the paper are sound, i.e., dictionary learning and voxel-wise encoding models. Their findings are also corroborated by related papers in the field, which also show strong correspondence between brain activity and LLMs (Schrimpf et al., 2021; Tuckute et al., 2023, Caucheteux et al., 2022). The fact that the learned dictionary response patterns can capture brain responses at all is interesting, given that the AN temporal response patterns are derived independent of brain activity.

**Weaknesses:**

My greatest concern is that the conceptual motivation for this manuscript appears confused and limited. The manuscript is largely motivated by understanding the functional organization of LLMs through the lens of the human brain. Understanding LLMs through the lens of the human brain assumes that we have a strong understanding of how the human brain works. But one can argue that our understanding of LLMs is actually stronger than our understanding of the human brain: We have access to all parameters, the objective function, and optimization procedures. Thus, it’s unclear what understanding is gained by understanding LLMs through the ‘functional organization of human brain networks’.

Suppose we give the manuscript the benefit of the doubt; perhaps there is something to be learned by evaluating whether LLMs exhibit the same functional organization as human brain networks. From the reported results, it appears that only language and visual networks appear to be represented. These findings seem limited, and have already been previously described (e.g., Kumar et al., 2024). Moreover, not all functional brain networks appear to be equally represented within the LLM, which is not entirely consistent with the title of the paper.

In terms of precedent, it’s also slightly unclear how the main findings of this paper conceptually differ from Kumar et al. (2024), which doesn’t appear to be cited.

Kumar, Sreejan, Theodore R. Sumers, Takateru Yamakoshi, Ariel Goldstein, Uri Hasson, Kenneth A. Norman, Thomas L. Griffiths, Robert D. Hawkins, and Samuel A. Nastase. “Shared Functional Specialization in Transformer-Based Language Models and the Human Brain.” Nature Communications 15, no. 1 (June 29, 2024): 5523. https://doi.org/10.1038/s41467-024-49173-5.

**Questions:**

How does this paper differ from Kumar et al., (2024)?

How do the authors reconcile the primary finding that visual/language networks are represented in the LLMs with the claim of the paper: brain-like functional organization in LLMs? (The brain is not limited to just visual and language networks.)

The authors write that this approach distinguishes itself from a population-level approach by identifying sub-groups. What is the difference between a population-level and sub-group level?

More minor questions:

BERT seems to be an ill-chosen model for this comparison (human v LLM), given that it is bidirectional. In the human dataset, words are presented sequentially (unidirectionally). Yet with BERT, words are provided/accessed in parallel; past words have access (or attended) to embeddings from future words. Conceptually, it’s not clear to me why one would include this model for analysis/comparison with human data, since it’s an unrealistic scenario.

On line 370, they mention the observation that FBN involvement is sparser in Llama3, and is suggested that more advanced LLM models may promote more compact brain-like functional organizations. Does this analysis account/control for LLM size?

What number of atoms were used when learning their dictionary? Do the results depend on the choice of k?

---

### Official Review · Reviewer_mNZN · 2024-11-02

**Soundness:** 2
**Presentation:** 2
**Contribution:** 2
**Rating:** 5
**Confidence:** 4

**Summary:**

This work focuses on a important research question, are there similarities in the organization of LLMs and the brain. This question is of interest to both the ML and neuroscience community, as insights into this question will likely help inform increased understanding of the brain as well as provide insights into how to build better ML models.

To address this question, this work proposes a novel approach to study the alignment between LLMs and brain activity as measured by fMRI. The new approach takes all artificial neurons (ANs) in a LLM and extracts representative patterns from the ANs. The novelty is that this approach enables the LLM representation to span across the LLMs layers, whereas prior approaches focused on layer specific representations.

This approach is applied to one session from the Narratives fMRI dataset, and 4 models (BERT, Llama 1-3), to begin to yield insights into the scientific quesetion. Some analysis of the output of applying this approach is then presented.

**Strengths:**

1. This work focuses on a important research question, are there similarities in the organization of LLMs and the brain. This question is of interest to both the ML and neuroscience community, as insights into this question will likely help inform increased understanding of the brain as well as provide insights into how to build better ML models.
2. This work proposes a novel approach to study the alignment between LLMs and brain activity as measured by fMRI. The new approach takes all artificial neurons (ANs) in a LLM and extracts representative patterns from the ANs. The novelty is that this approach enables the LLM representation to span across the LLMs layers, whereas prior approaches focused on layer specific representations.

**Weaknesses:**

Major:
1. Imprecise framing leads to the paper over promising what the paper presents:
(a) The paper proposes as a gap in the field that prior work was at a "population-level" and "individual ANs have yet to be linked to neural response". Despite this framing this paper doesn't fill this gap. Specifically, this work doesn't link individual ANs to neural response. Rather this work takes all ANs and extracts representative patterns from these ANs (dictionary) and then links these representative patterns to neural response (brain activity as measured by fMRI). While this is still a novel approach, the framing suggests that this paper links individual ANs to brain activity and the subsequent work presented does not accomplish this.
(b) Additionally, the abstract claims that "This research represent the first exploration of brain-like functional organization within LLMs". It is true that this is the first time this specific approach has been proposed, however it is easy to argue that the multiple prior works on alignment between LLMs and "neural responses" were also focused on understanding brain-like functional organization within LLMs.

2. While the approach is novel it is unclear from the work presented if it is simply a new approach, or if the scientific insights are complementary to prior approaches which align LLM layer representations to brain activity as measured by fMRI. As the goal of both types of approaches is to study the similarities between LLMs and the brain it would be valuable to highlight if this novel approach leads to new scientific insights.

3. Missing Key Technical Details:
(a) Multiple hyperparameters are mentioned, however there are no details shared about how these hyperparameters were selected. This includes the dictionary size (k), sparsity constraint (lambda_AN), and lambda_f. Especially as there is a clear decrease in the performance of dictionary learning for the larger models (Fig. 2 a) it is important to understand why the specific dictionary size, as opposed to a larger dictionary size was chosen.
(b) Details on how the voxel-wise encoding models were trained are missing, for example was cross-validation used, how large were the folds, etc. These are crucial details to be able to replicate the work and judge the validity of the learned models.

4. Missing Key Experimental Details:
(a) Details on how the FBNs that were chosen are missing. There are many networks defined in many different ways throughout the computational neuroscience literature, therefore this choice can have a big impact on scientific insights from the approach presented as well as provide insights into how generalizable this new approach is.
(b) Limited details about the Narratives dataset are presented. For example it is unclear how many TRs exist, the length of the session used, whether the subjects were instructed to stare at something during the experiment, etc.

5. Low R^2 for encoding models: Despite the paper stating that the R^2 range in Figure 2b "closely resembles" scores from prior studies, these R^2 values are actually low relative to what is common from encoding models predicting fMRI activity for naturalistic stimuli. This suggests that the representative patterns from these ANs is not as good of a predictor from fMRI activity as other representations. If low R^2's are common for this specific dataset and selected fMRI session selected, it would be beneficial to apply the proposed approach to another dataset or session to verify that the low predictive ability was due to the data as opposed to the approach. It would also be helpful to reference (if they exist) the explicit ranges of other encoding models using the same dataset and session.

6. Overly strong take aways based off limited results: There are multiple suggestions/claims that are made without considering alternative explanations or explaining potential bias based on the limited results presented. For example, in section 4.3 there is a focus on the average correlation coefficient over all possible atom pairs 6f, the results lead to a take away about Llama3. However given the limited number of results the take away is based on, an alternative explanation is that there are fewer (only 3) atoms for the Llama3 example as opposed to 4-6 atoms for the other models and given this it's possible that the higher correlation is simply due to having fewer atoms to measure correlation between leading to less variability.

Minor:
1. There are some typos throughout the paper, including (a) page 5 section 4.1 "vaules" is likely intended to be "values", (b) page 6 section 4.2 "involvement of FBNs...across different FBNs" it is likely that the second FBN was meant to refer to something else, (c) page 7 section 4.2 mentions "studies" but only cites one study Yeshurun et al. 2021.

2. At times the paper is unclear especially in how the figures are presented and described in the text. For example in Figure 2 the audience is left to interpret what probability (the y-axis) is in reference to. Also in figure 2 the corresponding part of the figure is sometimes before the description and sometimes after the description.

**Questions:**

1. The abstract and introduction should be reframed to prevent the work from over promising both in terms of novelty, and over promising in terms of the granularity of the analysis. See weaknesses Major 1 for more context.
2. Suggestions/take aways based on experiments should be reframed to clarify that these are just one plausible interpretation based on the results, and either alternatives should be described as well or limitations should be clearly highlighted which impact the interpretation of the results. One example is provided in weakness Major 6.
3. How do the insights from this approach differ from insights from prior approaches which align LLM layer representations to brain activity as measured by fMRI?
4. How were the hyperparemeters referenced in the paper such as dictionary size (k), sparsity constraint (lambda_AN), and lambda_f chosen?
5. How were voxel-wise encoding models were trained, see weakness major 3b for some examples of the missing information?
6. What are the missing key experimental details highlighted in weaknesses major 4b?
7. Additional details should be provided to explain the low R^2 for the encoding models in Fig 2b, whether that is a reference to explicit ranges of other encoding models using the same dataset and session, or applying the proposed approach to another dataset or session to verify that the low predictive ability was due to the data as opposed to the approach.

---

### Official Review · Reviewer_EPVL · 2024-11-03

**Soundness:** 3
**Presentation:** 3
**Contribution:** 3
**Rating:** 6
**Confidence:** 2

**Summary:**

This paper aims to compare the functional organization of large language models (LLMs) with the brain. It measures this correspondence using sub-groups of artificial neurons (ANs) in LLMs, established functional brain networks (FBNs) in the literature, and Narratives—a publicly available fMRI dataset.

In each of 4 popular LLMs, the authors extract the responses of all ANs to the tokenized input of the Narratives stimuli. These activations are then sparsely decomposed into representative ‘atoms,’ which are subsequently used in voxel-wise encoding model. This method yields predicitvity scores of fMRI responses in individual subjects.

Finally, the paper investigates the activity and organization of the representative atoms in each of the LLMs by correlating temporal responses and measures the spatial distribution of atoms across LLM layers.

Key to this paper’s claim is that most models comparing neural networks to the brain do so at a population level. The ‘atoms’ representing LLM activity are thus a way to represent individual and sub-groups of ANs and their correspondence to the brain.

Should the weaknesses described below be adequately addressed, I would weakly recommend accepting the paper.

**Strengths:**

Strengths:

Nicely written intro with a clear motivation and big question

Overall the sparse encoding model linking LLMs to the brains is an interesting and topical approach with relevance to on-going work in both machine learning and cognitive neuroscience.

Additionally, the focus on sub-population encoding models provides a novel approach in comparing LLMs to the brain.

**Weaknesses:**

Weaknesses:
The critical method used in this paper is the sparse decomposition of LLM responses into ‘atoms’ that represent sub-groups of ANs.  The parameters of this representation are stated clearly in lines 243-245, but there is no motivation provided for choosing these values. Why 64 atoms, what are the consequences of other numbers, and what goes into the choice of sparsity constraints? These questions should be acknowledged.

The paper claims to be the first study to characterize functional brain-like organization in LLMs, yet similar work (eg. Schrimpf et al, 2021) is acknowledged in the paper, apparently contradicting this claim. Instead, perhaps the claim should focus on characterization using sparse, sub-population level methods.

Automatic brain labeling (Kong el al, 2024) is a central method to this paper’s findings, but how this value is calculated is not mentioned. This method should be motivated and explained in more detail.

The brain predictivity scores are modest (figure 2B) and do not seem comparable to cited similar studies despite the authors’ claims (lines 265-268).

**Questions:**

I’m confused by figure 6. The second column seems to compare temporal consistency amongst atoms, but the diagonal (measuring the consistency of the same atom) appears to be 0. Should these values not be the highest, and—if so—why are they colored green? I also am unsure what the third column to meant to convey. They plots seem to be histograms of ANs across layers: should the values all be small (due to the sparsity constraint), and how does this relate to the functional organization of the LLM and its correspondence to the brain?

Figure 4 shows the brain maps associated with different FBNs. What is the criteria for a brain map to be classified as ‘associated’?

---

### Official Review · Reviewer_WbQS · 2024-11-04

**Soundness:** 1
**Presentation:** 3
**Contribution:** 1
**Rating:** 1
**Confidence:** 4

**Summary:**

This paper utilizes a sparse representation method to extract representative response patterns from multiple LLMs. These are then used to create voxel-wise encoding models of fMRI data. The performance of these models is used to link functional brain networks to the response patterns in the LLMs.

**Strengths:**

1. The method is clearly presented.
2. The paper compares scores with prior research to ensure consistency.

**Weaknesses:**

1. It is unclear why the whole brain was used in this analysis especially when the dataset itself has no inter-subject correlation for most of the brain (Fig 4 in Nastase et al. 2021). This is including the lateral visual cortex which is heavily discussed within the paper (Fig 3. among others).
2. I am unsure how this encoding model design can give insight into how LLM's are functionally organized. This is a regression approach in which the atoms can both linearly encode both some aspects of the brain and the LLM. Already the r^2 values for predicting brain response is 0.1 at max (fig. 2). Along with the values for the LLM being .5-.6 it is unclear what exactly is being captured.
3. It is unclear how the data is aggregated across fMRI subjects or if it is at all. Furthermore the only measure of consistency is the number of atoms that are active/deactive for each FBN (Fig 4). No inter-subject comparison is shown. Fig 6 e and f also show fairly low temporal consistency of atoms.
4. "offering novel insights to inform the development of artificial general intelligence (AGI) with human brain principles." There is no mention for how this method can lead to different approaches in building LLM's.




Small corrections:
Line 258 wrong spelling: "vaules"

**Questions:**

1. Why is there such a strong activation/deactivation of the lateral visual cortex for fMRI data with no visual component?
2. How was data aggregated across participants?

---

### Note · Authors · 2024-11-29

I have read and agree with the venue's withdrawal policy on behalf of myself and my co-authors.